# Defining key roles for auxiliary proteins in an ABC transporter that maintains bacterial outer membrane lipid asymmetry

Shuhua Thong[1†], Bilge Ercan[1†], Federico Torta[2], Zhen Yang Fong[1], Hui Yi Alvina Wong[1], Markus R Wenk[2], Shu-Sin Chng[1,3*]

[1]Department of Chemistry, National University of Singapore, Singapore, Singapore; [2]Department of Biochemistry, National University of Singapore, Singapore, Singapore; [3]Singapore Center on Environmental Life Sciences Engineering, National University of Singapore, Singapore, Singapore

*For correspondence: chmchngs@nus.edu.sg

[†]These authors contributed equally to this work

Competing interests: The authors declare that no competing interests exist.

**Abstract** In Gram-negative bacteria, lipid asymmetry is critical for the function of the outer membrane (OM) as a selective permeability barrier, but how it is established and maintained is poorly understood. Here, we characterize a non-canonical ATP-binding cassette (ABC) transporter in *Escherichia coli* that provides energy for maintaining OM lipid asymmetry via the transport of aberrantly localized phospholipids (PLs) from the OM to the inner membrane (IM). We establish that the transporter comprises canonical components, MlaF and MlaE, and auxiliary proteins, MlaD and MlaB, of previously unknown functions. We further demonstrate that MlaD forms extremely stable hexamers within the complex, functions in substrate binding with strong affinity for PLs, and modulates ATP hydrolytic activity. In addition, MlaB plays critical roles in both the assembly and activity of the transporter. Our work provides mechanistic insights into how the MlaFEDB complex participates in ensuring active retrograde PL transport to maintain OM lipid asymmetry.

## Introduction

The cell envelope of Gram-negative bacteria such as *Escherichia coli* is composed of two lipid bilayers termed the inner and outer membranes. The presence of the outer membrane (OM) makes Gram-negative bacteria generally resistant to external insults, including antibiotics and detergents, and allows these bacteria to survive in harsh environments (*Nikaido, 2003*). Unlike the inner membrane (IM), which is a phospholipid (PL) bilayer, the OM contains lipopolysaccharides (LPS) and PLs in the outer and inner leaflets, respectively (*Kamio and Nikaido, 1976*). This unique lipid asymmetry, characterized by a tightly-packed LPS outer leaflet, renders the OM impermeable to a wide range of compounds, including hydrophobic molecules (*Nikaido, 2003*).

During growth or in the event of stress, PLs may appear in the outer leaflet of the OM (*Jia et al., 2004*; *Wu et al., 2006*; *Dalebroux et al., 2014*). This disrupts the LPS layer and reduces the barrier function of the OM. The cell has developed mechanisms to cope with such perturbations on lipid asymmetry. Two OM β-barrel enzymes degrade PLs that have accumulated in the outer leaflet of the OM; OmpLA cleaves both acyl chains from the glycerol backbone of PLs (*Dekker, 2000*) while PagP transfers one acyl chain from PLs to LPS (*Bishop, 2005*) or phosphatidylglycerol (PG) (*Dalebroux et al., 2014*). In addition, the OmpC-Mla system is thought to maintain OM lipid asymmetry by removing PLs from the OM and transporting them back to the IM (*Malinverni and Silhavy, 2009*; *Chong et al., 2015*). Osmoporin OmpC forms a complex with the OM lipoprotein MlaA that

**eLife digest** *Escherichia coli* are bacteria that can cause vomiting and diarrhoea in humans and other mammals. Each *E. coli* cell is surrounded by two membranes, which are each made of two layers of fat molecules known as lipids. The outer membrane prevents the entry of toxic compounds and allows *E. coli* to withstand damaging agents from outside the cell, such as antibiotics.

The outer membrane's ability to act as an effective barrier depends on an asymmetric, or uneven, distribution of lipid molecules across its two layers. The inside layer is dominated by phospholipids, whereas the outside layer is comprised mainly of lipids with attached sugars. The distribution of the two lipid types is maintained by a molecular machine with components that can be found in both the inner and outer membranes. This machine is thought to remove phospholipids from the outside layer of the outer membrane and transport them back to the inner membrane.

A group (or"complex") of proteins known as MlaFEDB operates as a part of this machine at the inner membrane. MlaFEDB is believed to use energy derived from the breakdown of a molecule called ATP to help ensure that phospholipids removed from the outside layer of the outer membrane are reinserted into the inner membrane. It was proposed that the complex contains four proteins, but it was not clear exactly how these components are arranged. Now, Thong et al. reveal how MlaFEDB is organized and characterize the roles of the individual protein components.

The experiments confirm that the MlaFEDB complex is made up of four proteins, including two core components and two support proteins (called MlaB and MlaD). There are six copies of MlaD in the complex. In addition, MlaD has a strong affinity for phospholipids and plays a role in controlling the rate at which energy is harnessed through the breakdown of ATP.

Further experiments show that the other support protein MlaB is necessary for both the proper assembly and activity of the complex, likely through its interaction with one of the core components. The next step following on from this work is to directly observe MlaFEDB in action to find out how it uses energy to insert lipids into the inner membrane. In the long term, more information about the structure of the complex would be needed to further understand how it works at the molecular level.

likely removes PLs from the outer leaflet of the OM (*Chong et al., 2015*). The periplasmic chaperone MlaC is believed to transport these extracted PLs across the aqueous periplasm, and hand them over to a putative ATP-binding cassette (ABC) family transporter, MlaFEDB, at the IM (*Malinverni and Silhavy, 2009*). Whether these PLs get inserted into the outer leaflet or transported back to the inner leaflet of the IM is not known.

The exact composition and the roles of the respective components of the ABC transporter complex have not been elucidated. MlaE and MlaF constitute the core components of the ABC transporter and are predicted to form the transmembrane domains (TMDs) and nucleotide-binding domains (NBDs), respectively (*Malinverni and Silhavy, 2009*). In addition to MlaC, this system is proposed to contain a second substrate-binding protein (SBP), a single-pass membrane protein MlaD, that may be associated with the IM complex. Consistent with this idea, MlaD, along with MlaE and MlaF, are conserved across many species of Gram-negative bacteria and can also be found in actinomycetes (*Casali and Riley, 2007*) and in plants (*Benning, 2009*). The Mce4 pathway in *Mycobacterium tuberculosis* is important for cholesterol uptake (*Pandey and Sassetti, 2008*) while the TGD system in the chloroplasts of *Arabidopsis thaliana* functions to transport phosphatidic acid (PA) from the plastid OM to the IM (*Benning, 2009*), providing support for the proposed role of the OmpC-Mla system in lipid transport. In *E. coli* and a few other Gram-negative bacteria, a small cytoplasmic protein MlaB is also predicted to be part of the ABC transporter (*Malinverni and Silhavy, 2009*; *Casali and Riley, 2007*). MlaB contains a Sulfate Transporter and Anti-Sigma factor antagonist (STAS) domain that is believed to have a general nucleoside triphosphate (NTP) binding function (*Aravind and Koonin, 2000*). Its role in the IM complex is unclear.

Here, we characterize the putative MlaFEDB complex in *E. coli*, and elucidate critical roles for auxiliary proteins, MlaD and MlaB, in the assembly and activity of the complex. We show that MlaF, MlaE, MlaD and MlaB interact specifically with each other to form a stable complex. Within this

complex, MlaD forms an SDS-resistant hexamer via its soluble domain, and we demonstrate that this domain binds PLs. We further show that MlaD depresses ATP hydrolytic activity upon association with the MlaFEB sub-complex. Finally, we establish that MlaB is necessary for both the assembly and activity of the ABC transporter, likely by modulating MlaF structure and stability. Our work provides novel mechanistic insights into how the MlaFEDB complex functions to maintain lipid asymmetry in the OM.

## Results

### MlaF, MlaE, MlaD and MlaB form a stable complex

To determine whether MlaF, MlaE, MlaD and MlaB interact with each other, we performed affinity purification experiments using wild-type (WT) cells exogenously expressing N-terminally His-tagged MlaE protein (His-MlaE). His-MlaE is expressed from a "leaky" expression plasmid that has been shown to yield low cellular levels of other proteins (*Wu et al., 2006*; *Chong et al., 2015*). This construct is fully functional, as it is able to restore SDS/EDTA resistance in the Δ*mlaE* mutant strain (*Figure 1—figure supplement 1*). Two unique protein bands at ~29 kDa and ~19 kDa co-purified with His-MlaE (*Figure 1A*). These bands represent MlaF and MlaD, respectively, as they are no longer co-purified in the corresponding Δ*mlaF* and Δ*mlaD* mutant strains. Even though His-MlaE enabled enrichment of MlaF and MlaD, we are unable to detect the His-tagged protein in these samples, which have been heated prior to analysis. As there is a tendency for hydrophobic membrane proteins to aggregate when heated, we wondered if this was true for His-MlaE. Indeed, we are able to detect His-MlaE on immunoblots when the samples are not heated (*Figure 1A*, right panel). His-MlaE migrates anomalously as a diffuse band around 24 kDa (expected 28 kDa), likely because it is still partially folded in the presence of SDS, a phenomenon commonly observed for multi-pass membrane proteins (*Rath et al., 2009*). Remarkably, MlaD also migrates differently, now as a high molecular weight species, when the samples are not heated, suggesting that it may be forming oligomers. To examine if the interactions between MlaF, MlaE and MlaD were specific, we performed reciprocal affinity purification experiments using WT cells expressing C-terminally His-tagged MlaF (MlaF-His) or MlaD (MlaD-His) proteins, which we show are functional (*Figure 1—figure supplement 1*). Even though we could not probe for MlaE due to the lack of appropriate antibodies, we demonstrate that MlaF-His and MlaD-His are able to pull down MlaD and MlaF, respectively, likely via interactions with MlaE (*Figure 1—figure supplement 2A*). Taken together, these results establish that MlaF, MlaE and MlaD interact specifically in a complex.

We were not able to detect MlaB in our pulldown experiments, suggesting either that it is not part of the complex or it may have eluded detection due to its small size (~10 kDa). To determine whether MlaF, MlaE and MlaD form a stable complex with MlaB, we over-expressed the *mlaFEDCB* operon engineered to encode His-MlaE, and purified the complex on cobalt affinity resin followed by size exclusion chromatography (SEC). The proteins elute as a single peak, indicating the formation of a stable complex (*Figure 1B*). SDS-PAGE analysis of the complex show three bands at ~29 kDa (MlaF), ~20 kDa (MlaD) and ~10 kDa (MlaB) when the sample is heated. The identities of these bands are confirmed by mass spectrometry (MS) (*Figure 1—figure supplement 3*). In the non-heated sample, we are also able to detect His-MlaE (~24 kDa) and the oligomeric form of MlaD (~120 kDa) (*Figure 1B*). Consistent with MlaB being a part of the complex, we are able to show that functional His-tagged MlaB can pull down the other components in WT cells (*Figure 1—figure supplements 1* and *2B*). Even though we also expressed MlaC, it was not co-purified with the rest of the complex. This suggests that interactions between MlaC and the IM complex may be weak and/or transient, in line with the idea that it is a periplasmic binding protein (*Malinverni and Silhavy, 2009*). We conclude that MlaF, MlaE, MlaD and MlaB form a stable complex.

### MlaD alone exists in an extremely stable hexameric form

MlaD appears to form an oligomeric structure stable to SDS when purified as part of the IM complex. Full length MlaD comprises an N-terminal transmembrane helix and a large C-terminal periplasmic substrate-binding domain (*Figure 2A*) (UniProtKB P64604) (*Magrane and the UniProt consortium, 2011*). To determine whether this oligomeric state is dependent on the transmembrane helix or its association with the complex, we over-expressed and purified the substrate-binding

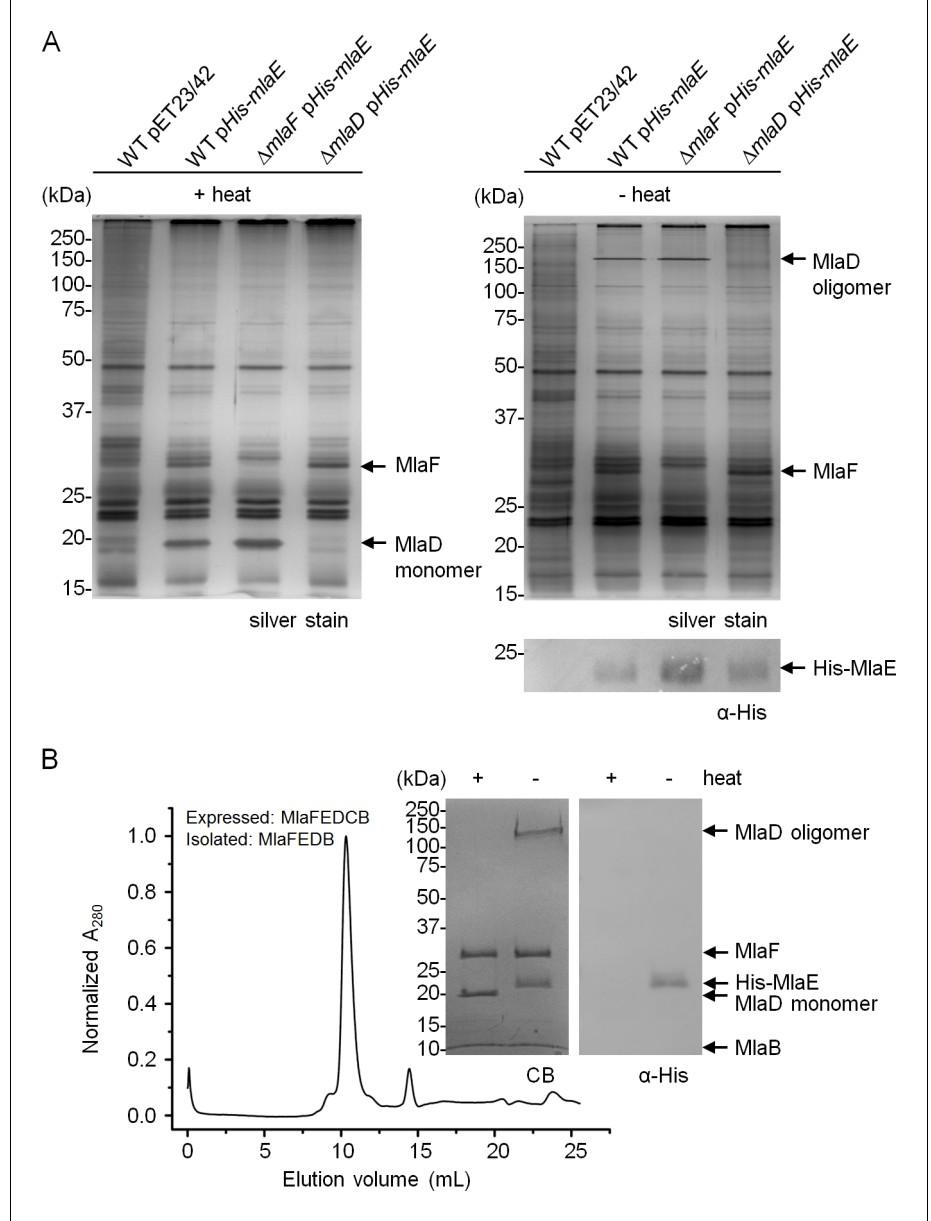

**Figure 1.** MlaF, MlaE, MlaD and MlaB form a stable complex. (**A**) Co-TALON affinity purification using WT and indicated mutant strains harboring empty vector (pET23/42) or pET23/42*His-mlaE* (p*His-mlaE*). Samples (heated or non-heated) were subjected to SDS-PAGE (12% Tris.HCl gel), and visualized by silver staining and immunoblot analyses using antibodies against the pentahistidine tag. (**B**) SEC profile of MlaF(His-E)DB complex purified from cells over-expressing MlaF(His-E)DCB. The peak fraction (heated or non-heated) was subjected to SDS-PAGE (4–20% Tris.HCl gel) followed by Coomassie Blue (CB) staining and immunoblot analysis. His-MlaE can only be detected on immunoblots when samples are not heated. Under the same conditions, MlaD migrates as a high molecular weight species. Positions of relevant molecular weight markers are indicated in kDa.

The following figure supplements are available for figure 1:

**Figure supplement 1.** His-tagged Mla proteins are able to rescue SDS/EDTA sensitivity in the respective *mla* mutant strains.

**Figure supplement 2.** MlaF, MlaE, MlaD and MlaB form a stable complex.

**Figure supplement 3.** MlaF, MlaD and MlaB co-purify with His-tagged MlaE following overexpression and affinity purification.

*Figure 1 continued on next page*

*Figure 1 continued*

**Figure supplement 4.** SEC-MALS analysis of the MlaF(His-E)DB complex.

domain (or soluble domain) of MlaD (sMlaD-His) alone for characterization. We found that sMlaD-His does not exist in the monomeric state, as judged by its SEC profile (*Figure 2B*). Multi-angle light scattering (MALS) analysis revealed that the absolute molar mass of purified sMlaD-His is ~110.7 kDa, establishing that these are in fact hexamers (*Figure 2C*). Furthermore, the experimental molar

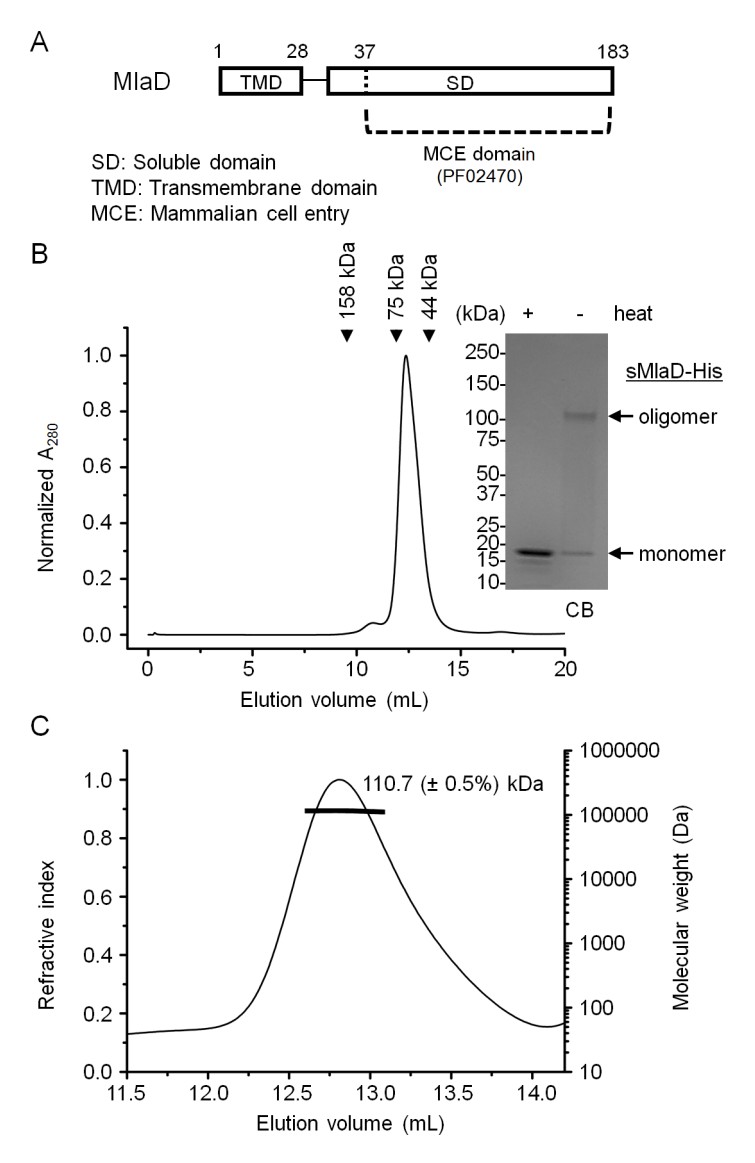

**Figure 2.** MlaD forms SDS-resistant hexamers via its soluble domain. (**A**) Domain organization of MlaD. (**B**) SEC profile of purified soluble domain of MlaD (sMlaD-His). Elution volumes of standard globular proteins (aldolase 158 kDa, conalbumin 75 kDa and ovalbumin 44 kDa) are indicated. The peak fraction (heated or non-heated) was subjected to SDS-PAGE (4–20% Tris.HCl gel) followed by CB staining. Positions of relevant molecular weight markers are indicated in kDa. (**C**) SEC-MALS analysis of sMlaD-His. Hexamer molecular mass: 107 kDa (predicted), 110.7 ( ± 0.5%) kDa (observed). Numbers stated after ± show statistical consistency of the analysis.

mass of the MlaFEDB complex (~285 kDa by SEC-MALS) is consistent with the presence of six copies of MlaD (*Figure 1—figure supplement 4*). Remarkably, hexamers formed by sMlaD-His alone are also resistant to SDS denaturation (*Figure 2B*). These results indicate that oligomerization and extreme stability are unique properties of the soluble domain.

## The soluble domain of MlaD binds endogenous phospholipids

The TGD pathway in *A. thaliana* has been proposed to transport PA between the inner and outer membranes of chloroplasts (*Benning, 2009*). In this system, TGD2 is the direct homolog of MlaD and was found to bind PA (*Awai et al., 2006*; *Roston et al., 2011*). We therefore hypothesized that MlaD also binds PLs as part of its role in maintaining lipid asymmetry in the OM. As a first step, we examined whether our purified preparations of sMlaD-His contained bound PLs. We extracted potentially bound PLs from purified sMlaD-His and analyzed these samples using thin layer chromatography (TLC) and $^{31}$P nuclear magnetic resonance (NMR) spectroscopy. We use delipidated LolB (dLolB-His), the OM receptor for OM-targeted lipoproteins, as a specificity control. LolB is not expected to bind PLs even though it contains a hydrophobic cavity that allows it to chaperone the triacyl modification found on lipoproteins (*Takeda et al., 2003*). We detected both phosphatidylethanolamine (PE) and PG bound to sMlaD-His (*Figure 3A and B*) but not to dLolB-His (*Figure 3A*), indicating that MlaD has affinity for PLs. Since sMlaD-His no longer contains the transmembrane helix and has little tendency to associate with membranes, we believe that bound PLs in sMlaD-His are directly reflective of its proposed role in PL transport. Comparable amounts of PE and PG (~1:1) co-purified with sMlaD-His despite PE being the predominant lipid in *E. coli* cellular extracts (~72% PE, 17% PG, 11% cardiolipin (CL)) (*Lugtenberg and Peters, 1976*) (*Figure 3B*). These results suggest that MlaD may have a preference for binding PG.

To further characterize the interaction between MlaD and PLs, we analyzed purified sMlaD-His using MS. In denaturing MS, the monomeric form of sMlaD-His (deconvoluted mass 17,828 Da) is the only molecular species observed (*Figure 3—figure supplement 2A*). In native MS, we detected the hexameric form of sMlaD-His with bound ligands at various charge states (*Figure 3C*). Deconvolution of the native MS spectrum indicates the molecular weight of the native hexamers to be ~110 kDa, suggesting the presence of at least four bound PL molecules (~750 Da each, in agreement with the average mass of PE and PG) and therefore, strong protein-PL interactions. By increasing the collision energy in the mass spectrometer collision cell, we are then able to destabilize the native structure, revealing the presence of hexamers binding three, two, one or zero PL molecules (*Figure 3—figure supplement 2B*). This suggests that the presence of ligands is not strictly required for the formation of sMlaD hexamers. Taken together, our data establish that hexameric sMlaD is a PL-binding complex with at least four binding sites.

## MlaB is required for the assembly of the IM ABC transporter

MlaE and MlaF constitute TMDs and NBDs of a canonical ABC transporter, respectively (*Malinverni and Silhavy, 2009*). MlaD binds PLs, consistent with its proposed role as a periplasmic binding protein; however, it is not clear if MlaD has other roles in the activity of the ABC transporter. The function of MlaB is also unknown. To characterize the importance of MlaD and MlaB in the complex, we attempted to over-express and purify sub-complexes containing His-MlaE, including MlaFE, MlaFED and MlaFEB (*Figure 4*). We were successful in obtaining purified MlaFEB (*Figure 4C*), but not MlaFE and MlaFED (*Figure 4A and B*). Despite reasonable expression of MlaF (*Figure 4—figure supplement 1*), it is not co-purified with His-MlaE in preparations of MlaFE and MlaFED (*Figure 4A and B*). Therefore, MlaF does not interact with MlaE unless MlaB is present. To exclude artifacts due to over-expression, we also performed affinity purification using His-MlaE expressed at low levels in cells lacking MlaB. Unlike in WT cells, MlaF is not co-purified with His-MlaE in the absence of MlaB (*Figure 4D*). Similarly, we show that MlaF-His could not pull down MlaD (through MlaE) without MlaB (*Figure 4—figure supplement 2*). In fact, the affinity-enriched levels of MlaF-His appear to be reduced in the absence of MlaB, suggesting that MlaB modulates the stability of MlaF. Our results demonstrate that MlaB plays a key role in the assembly of the complex.

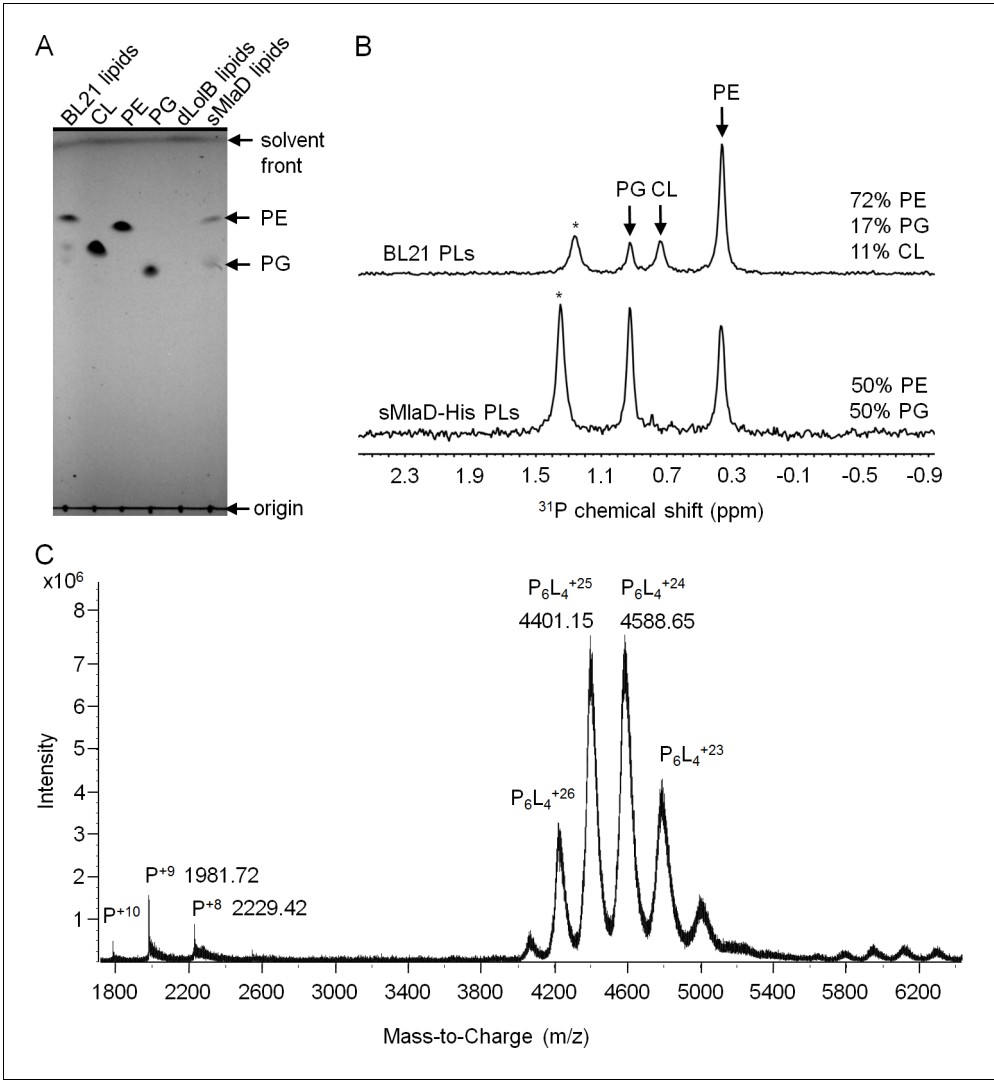

**Figure 3.** sMlaD co-purifies with endogenous PLs. (**A**) TLC analysis of PLs extracted from BL21(λDE3) cells, purified dLolB-His and sMlaD-His. (**B**) $^{31}$P NMR analysis of PL extracts from BL21(λDE3) cells and purified sMlaD-His in 5% Triton X-100. Compositions of bound PLs were obtained via integration of peak areas, and normalized to the number of phosphorus atoms per PL molecule (i.e. one for PE/PG and two for CL). Unknown peaks that cannot be assigned to any PL species in *E. coli* (see *Figure 3—figure supplement 1*) are annotated with asterisks (*). (**C**) Positive mode, non-denaturing electrospray ionization (ESI) mass spectrum of sMlaD-His. Under native conditions (20 mM ammonium acetate, pH 6.9), sMlaD hexamers with charge states centred around +24 could be detected. After deconvolution, the molecular weight of the native hexamers was ~110 kDa, indicating the presence of at least four bound PLs (i.e. $P_6L_4$, assuming an average mass of 750 Da per PL).

The following figure supplements are available for figure 3:

**Figure supplement 1.** $^{31}$P NMR analysis of *E. coli* PLs. $^{31}$P chemical shifts (ppm) of phosphatidic acid (14:0) (PA), phosphatidylserine (PS), cardiolipin (CL), phosphatidylglycerol (PG), and phosphatidylethanolamine (PE) are given in the table.

**Figure supplement 2.** MS analyses of sMlaD-His.

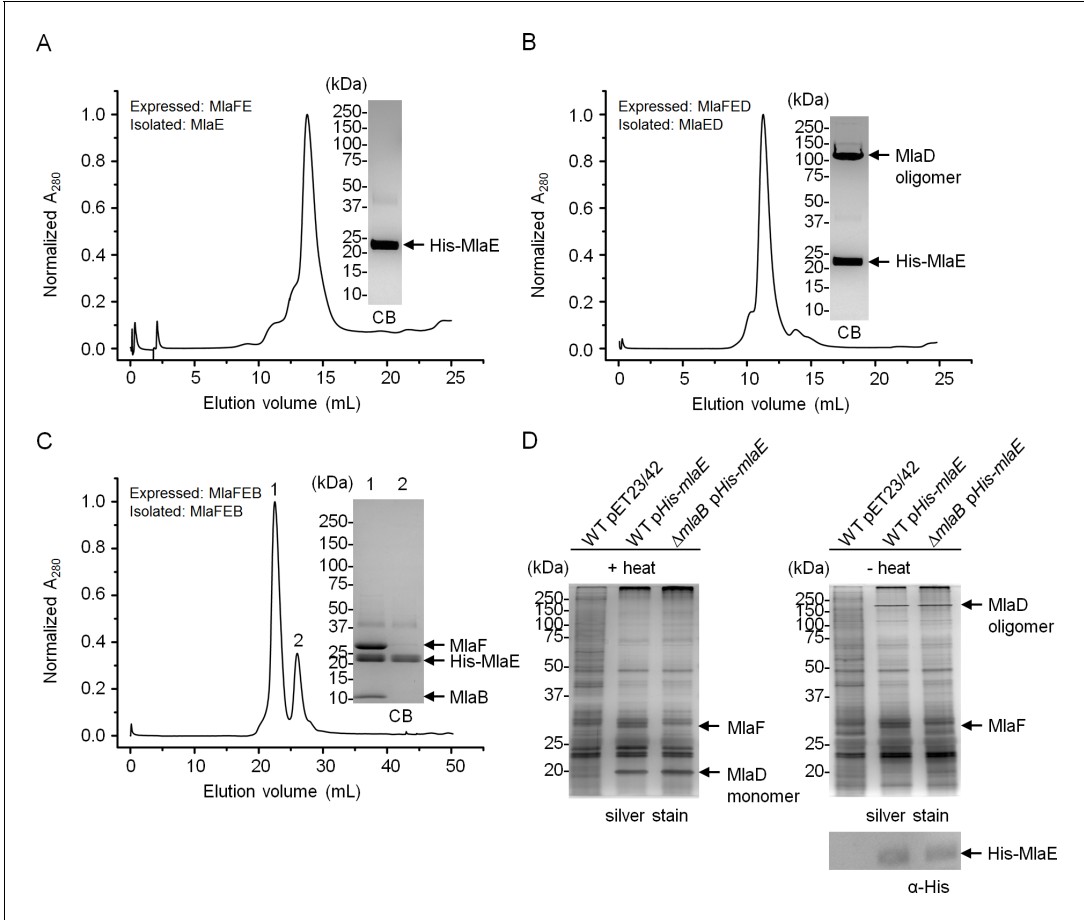

**Figure 4.** MlaB is required for the stability and/or assembly of the canonical ABC transporter. SEC profiles of (**A**) His-MlaE purified from cells over-expressing MlaF(His-E), (**B**) (His-MlaE)D purified from cells over-expressing MlaF(His-E)D, and (**C**) MlaF(His-MlaE)B purified from cells over-expressing MlaF(His-E)B. The respective peak fractions (non-heated) were subjected to SDS-PAGE (4–20% Tris.HCl gel) followed by CB staining. (**D**) Co-TALON affinity purification using WT and ΔmlaB strains harboring empty vector (pET23/42) or pET23/42His-mlaE (pHis-mlaE). Samples (heated or non-heated) were subjected to SDS-PAGE (12% Tris.HCl gel), and visualized by silver staining and immunoblot analyses using antibodies against the pentahistidine tag. Positions of relevant molecular weight markers are indicated in kDa.

The following figure supplements are available for figure 4:

**Figure supplement 1.** MlaF is produced at high levels in strains over-expressing full and sub-complexes of the IM ABC transporter.

**Figure supplement 2.** MlaD is not co-purified with MlaF-His in the absence of MlaB.

## Both MlaD and MlaB are important for the ATP hydrolytic activity of the IM ABC transporter

We next examined whether MlaB plays additional roles in the activity of the ABC transporter. To test this, we constructed a variant with a mutation in the STAS domain of MlaB at position 52, which in two other STAS domain proteins have been shown to be important for function (*Aravind and Koonin, 2000*; *Diederich et al., 1994*; *Rouached et al., 2005*). This mutation does not affect the assembly of the MlaFEB complex (*Figure 5A*), allowing us to compare the steady-state ATP hydrolysis rates of purified MlaFEB$_{WT}$ and MlaFEB$_{T52A}$ complexes. We demonstrate that MlaFEB$_{WT}$ exhibits high intrinsic ATPase activity in detergent micelles ($k_{cat}$ = 1.8 ± 0.5 µmol ATP s$^{-1}$/µmol complex) (*Figure 5A*), comparable to known ABC importers (*Reich-Slotky et al., 2000*; *Tal et al., 2013*). In addition, ATP binding is cooperative (Hill coefficient = 1.5 ± 0.5). Remarkably, no activity is detected with the MlaFEB$_{T52A}$ complex, similar to a variant of the complex containing a predicted non-

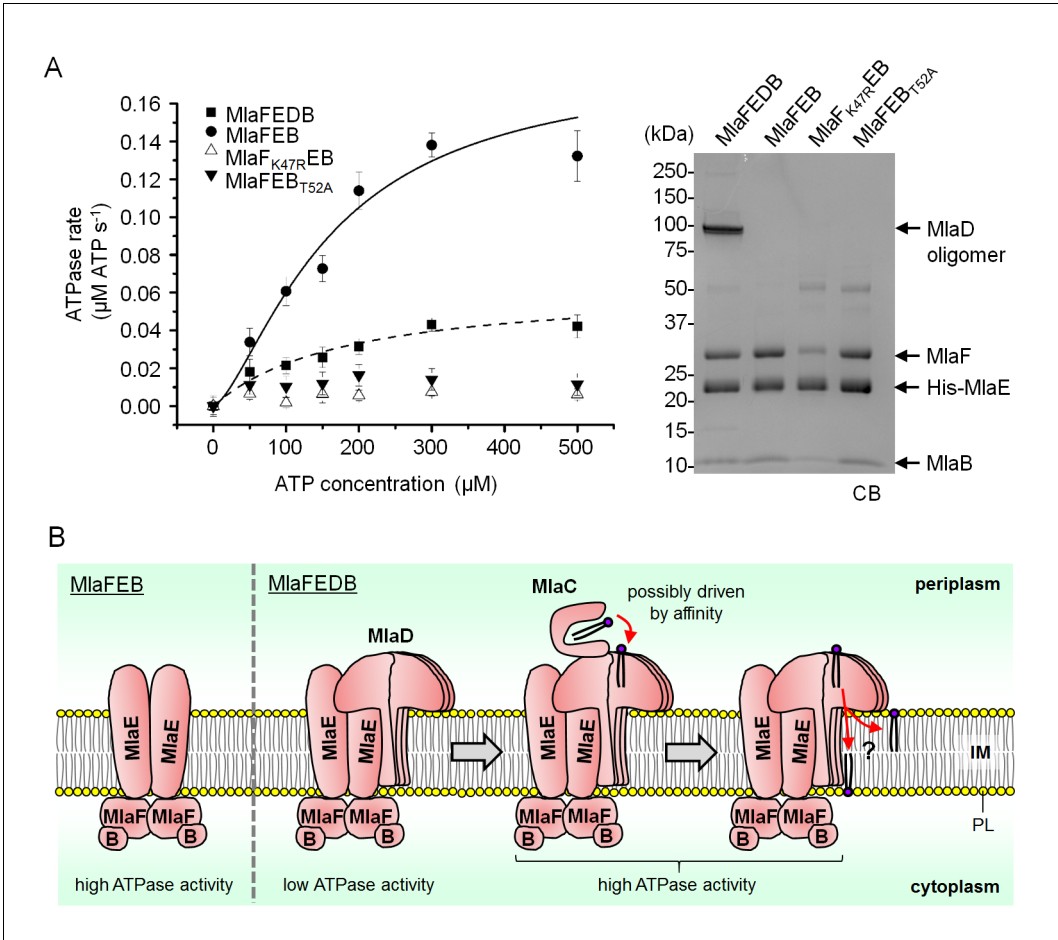

**Figure 5.** MlaD and MlaB modulate ATP hydrolytic activity of the IM ABC transporter. (**A**) Enzyme-coupled ATPase assays of indicated complexes (0.1 μM) performed in detergent micelles (0.05% DDM). Average ATP hydrolysis rates (obtained from triplicate experiments, see *Figure 5—source data 1*) were plotted against ATP concentrations, and fitted to an expanded Michaelis-Menten equation that includes a term for Hill coefficient (n); MlaFEDB ($k_{cat}$ = $V_{max}$/[complex] = 0.6 ± 0.3 μmol ATP s$^{-1}$/μmol complex, $K_m$ = 181.1 ± 203.6 μM, n = 1.0 ± 0.6) and MlaFEB ($k_{cat}$ = 1.8 ± 0.5 μmol ATP s$^{-1}$/μmol complex, $K_m$ = 161.4 ± 75.57 μM, n = 1.5 ± 0.5). SDS-PAGE analysis of the complexes (non-heated) used for these assays is shown on the right. Error bars in the graph and numbers stated after ± are standard deviations of triplicate data. (**B**) A proposed model for how the MlaFEDB complex functions to drive PL transport from the OM to the IM.

The following source data and figure supplement are available for figure 5:

**Source data 1.** Source data for ATPase assay.

**Figure supplement 1.** *mlaF_{K47R}* and *mlaB_{T52A}* are non-functional alleles.

functional mutation in the MlaF Walker A motif (MlaF$_{K47R}$EB) (*Walker et al., 1982*). Accordingly, the *mlaB$_{T52A}$* or *mlaF$_{K47R}$* alleles do not complement SDS-EDTA sensitivity observed in Δ*mlaB* and Δ*mlaF* strains, respectively (*Figure 5—figure supplement 1*). These results imply that MlaB assumes critical role(s) during catalysis, and validate this STAS domain protein as an essential and functional component of the IM ABC transporter.

SBPs are also known to modulate the activities of their associated ABC transporters (*Davidson et al., 2008*), so we asked whether the presence of MlaD affected the activity of the Mla-FEB complex. We show that the association of MlaD with MlaFEB does not significantly affect the affinity of the complex for ATP ($K_m$ = 181.1 ± 203.6 μM for MlaFEDB vs 161.4 ± 75.57 μM for

MlaFEB) (*Figure 5A*); however, its presence in the MlaFEDB complex reduces ATP hydrolytic activity by ~3-fold ($k_{cat}$ = 0.6 ± 0.3 µmol ATP s$^{-1}$/µmol complex) and affects cooperativity (Hill coefficient = 1.0 ± 0.6). Given that the MlaED complex can be stably isolated (*Figures 1A* and *4B*), and that MlaD does not co-purify with MlaF-His in the absence of MlaE (*Figure 4—figure supplement 2*), it is likely that MlaD interacts with the ABC transporter solely via MlaE. Our findings suggest that such an interaction may cause structural changes in MlaF (the NBDs) and/or directly influence how the two NBDs come together to perform catalysis.

## Discussion

The OmpC-Mla system is believed to maintain OM lipid asymmetry by removing PLs from the outer leaflet of the OM (OmpC-MlaA complex), and shuttling these PLs across the periplasm (MlaC) to the IM, where they get handed off to a putative ABC transporter (*Malinverni and Silhavy, 2009*; *Chong et al., 2015*). In this study, we have characterized the structure and function of the IM transporter. We have shown that MlaF, MlaE, MlaD and MlaB constitute and form a stable complex. Within this complex, we have demonstrated that MlaD forms SDS-resistant hexamers, and co-purifies with PLs. We have also revealed that MlaD modulates the ATPase activity of the complex via its interaction with MlaE. Finally, we have established multiple roles for MlaB in both the assembly and activity of the ABC transporter, and have identified a single mutation in MlaB that separated these two functions.

Substrate import via ABC transporters typically involves delivery of the substrate from a periplasmic SBP to the IM complex, where ATP binding and hydrolysis are coupled to substrate translocation across the membrane (*Davidson et al., 2008*). For a few well-characterized transporters, ATP hydrolytic activity is intrinsically low and can only be activated upon the binding of the corresponding SBPs (*Chen et al., 2001*; *Liu et al., 1997*), an arrangement believed to reduce the futile hydrolysis of ATP in the absence of substrate (*Davidson et al., 2008*). In the OmpC-Mla system, a second SBP (MlaD) with unknown function is associated with the IM ABC transporter (MlaFEB), in addition to the periplasmic SBP (MlaC) (*Malinverni and Silhavy, 2009*). We have shown that MlaD does indeed bind PLs (*Figure 3*), the proposed substrates for this system, consistent with its role as an SBP. In addition, we have demonstrated that MlaFEB possesses high intrinsic ATPase activity that is attenuated by MlaD association (*Figure 5A*). Interactions with MlaD may stabilize conformations that are inactive in ATP hydrolysis. Furthermore, because our MlaFEDB preparations are solubilized in detergent micelles, which can disrupt lipid binding, it may be possible that MlaD does not have PLs bound. While this idea remains to be tested, the complex may be unstimulated without bound substrates, resulting in low ATPase activity (*Figure 5B*). Our results suggest a function for MlaD in ensuring that the full complex conserves ATP, and presumably retains the capacity to be activated only when MlaC binds and/or delivers the PL substrate.

How the OmpC-Mla pathway mediates retrograde PL transport from the OM back to the IM (against a concentration gradient) is not clear. One can posit that ATP hydrolysis by the MlaFEDB complex in the IM is directly coupled to the removal of PLs from the outer leaflet of the OM. Such a scenario would likely require physical interactions between the IM and OM complexes, as is the case for the LPS transport (Lpt) machinery (*Chng et al., 2010a*; *Freinkman et al., 2012*). However, the facts that the MlaC homolog from *Ralstonia solenacerum* can be purified and crystallized as a soluble complex with PE (Protein Data Bank ID: 2QGU; DOI: 10.2210/pdb2qgu/pdb), and that we could not detect strong interactions between MlaC and the OmpC-MlaA (*Chong et al., 2015*) or MlaFEDB complexes, indicate that Mla-mediated PL transport across the periplasm may proceed instead via a soluble intermediate. In this alternative scenario, we posit that PL transfer from the OmpC-MlaA complex to MlaC, and then to MlaD in the IM complex, is largely driven by affinity (*Figure 5B*). We have demonstrated that MlaD forms homo-hexamers (*Figure 2*), suggesting a maximum of six PL binding sites within the MlaFEDB complex. At least four of these binding sites have reasonably high affinity for PLs, thereby facilitating co-purification with MlaD and detection in native MS analysis (*Figure 3*). MlaD may have higher affinity for PL substrates than MlaC, driving overall PL transfer (*Figure 5B*). In the MlaFEDB complex, the energy derived from ATP hydrolysis may then be utilized to release bound PLs into the membrane, perhaps by inducing conformational changes that alter PL binding affinities within the MlaD oligomer. The fate of these PL substrates, whether being

translocated across the IM or simply released into the outer leaflet of the membrane, remains an intriguing question.

We believe that the hexameric architecture of MlaD, which contains the MCE domain (*Figure 2A*), is conserved in homologs of the Mla system found in plants and actinomycetes. In the chloroplasts of *A. thaliana,* the TGD pathway has a similar function in PL transport from the OM back to the IM (*Benning, 2009*). Here, the MlaD-like protein, TGD2, has been proposed to exist as an oligomer (*Roston et al., 2011*, *2012*). In *M. tuberculosis*, there are four Mce systems, of which Mce4 is essential for cholesterol uptake and catabolism (*Pandey and Sassetti, 2008*), while others may have functions in transporting other lipids (*Forrellad et al., 2014*). These systems are expressed from operons each comprising genes encoding two MlaE-like permeases and six MlaD-like (MCE) proteins (*Casali and Riley, 2007*). The organization of these operons suggests the assembly of ABC transporters containing a hetero-hexameric arrangement of MCE domains. This hexameric MCE structure may thus be a general architecture for binding and transporting various lipid substrates.

MlaD appears to be specific for PG over PE, the major PLs in *E. coli* (*Figure 3*); however, we do not know if other minor anionic PLs, such as PA and PS, are also substrates. The observed preference for PG may reflect a higher affinity of MlaD for this lipid simply during purification conditions. Alternatively, it may indicate specificity for PG during PL transport. Since the OmpC-Mla system is proposed to remove PLs from the outer leaflet of the OM so as to maintain lipid asymmetry, this finding suggests that either PG has a higher tendency to appear on the cell surface during normal growth, or removal of outer leaflet PG is more important to ensure OM stability. Consistent with these ideas, the OM enzyme PagP has been evolved to acylate outer leaflet PG (in addition to LPS), but not PE, as a mechanism to fortify the OM barrier (*Dalebroux et al., 2014*). It has also been reported that the OM has lower PG content than the IM in *E. coli* (*Lugtenberg and Peters, 1976*) and *Salmonella* Typhimurium (*Osborn et al., 1972*). While these observations are still controversial, an active PG-specific retrograde transport system could have utility in establishing the apparent differences in membrane PL compositions.

The OmpC-Mla system is unique in that the IM ABC transporter contains a cytoplasmic STAS domain auxiliary protein, MlaB, whose function is not known (*Malinverni and Silhavy, 2009*). We have shown that MlaB is critical for the assembly of the ABC transporter. In the absence of MlaB, the canonical ABC transporter complex (MlaFE) could not be isolated (*Figure 4A*), owing to the decreased stability of MlaF and/or weakened interactions between MlaE and MlaF. We have also established a separate role for MlaB in the activity of the complex; a single T52A mutation in the STAS domain protein does not affect complex assembly but abolishes ATPase activity (*Figure 5A*). Corroborating these conclusions, we note that STAS domains have similarly been reported to be critical for the assembly and activity of non-ABC family transporters, including the Sultr1;2 sulfate transporter in *A. thaliana* (*Shibagaki and Grossman, 2006*). In fact, MlaB-like and MlaE-like domains can be found in the same polypeptides in some organisms (Pfam accession number: PF13466) (*Finn et al., 2016*), further highlighting the importance of MlaB in these transporters. MlaB may regulate ATPase activity via novel mechanisms. The T52A mutation removes a conserved threonine residue in MlaB that is mapped to a serine in the STAS domain anti-sigma factor antagonist, SpoIIAA, in *Bacillus subtilis* (*Aravind and Koonin, 2000*). This residue can be phosphorylated by its target anti-sigma factor SpoIIAB, which disrupts the ability of SpoIIAA to bind SpoIIAB, presumably due to a conformational change (*Diederich et al., 1994*). By analogy, we hypothesize that MlaB may become phosphorylated in cells. Due to its influence on MlaF structure and stability, any conformational changes associated with this putative phosphorylation event might indirectly regulate MlaF and affect its function in ATP hydrolysis. Intriguingly, SpoIIAA has also been shown to exhibit NTP binding and hydrolytic activity (*Najafi et al., 1996*). This raises an alternative possibility that MlaB may be directly involved in ATP hydrolysis in conjunction with MlaF, perhaps with T52 serving as an alternative nucleophile. While such mechanistic involvement has not been reported, direct participation by an auxiliary protein in catalysis would necessitate a shift in our understanding of how ABC transporters can operate.

## Materials and methods

### Bacterial strains and growth conditions

All bacterial strains used in this study are summarized in *Supplementary file 1*. Strains MC4100 [*F⁻ araD139 Δ(argF-lac) U169 rpsL150 relA1 flbB5301 ptsF25 deoC1 ptsF25 thi*] (*Casadaban, 1976*) and BL21(λDE3) [*F⁻ ompT gal dcm lon hsdS$_B$*(r$_B^-$m$_B^-$) λ(DE3 [*lacI lacUV5-T7 gene 1 ind1 sam7 nin5*]) [*malB⁺*]$_{K-12}$(λ$^S$)] (Novagen) were used for in vivo and protein over-expression experiments, respectively. Deletion mutants were generated by P1 transduction using BW25113 [*F⁻ Δ(araD-araB)567 ΔlacZ4787::rrnB-3 λ⁻ rph-1 Δ(rhaD-rhaB)568 hsdR514*] mutant strains from the Keio collection (*Baba et al., 2006*). MlaB deletion mutation was constructed using recombineering as previously described (*Datsenko and Wanner, 2000*; *Malinverni and Silhavy, 2009*). Primers MlaB-cam-N5 and MlaB-cam-C3 (*Supplementary file 3*) with homologies to regions upstream and downstream of *mlaB*, were used to amplify the CamR cassette from pKD3 by PCR. The resulting PCR product was used to replace the *mlaB* gene by electroporation into BW25113 expressing λ$_{Red}$ recombinase (pKD46) (*Datsenko and Wanner, 2000*). The *mlaB::cam* allele was subsequently introduced into the wild-type (WT) strain by P1 transduction. Luria-Bertani (LB) broth and agar were prepared as previously described (*Silhavy et al., 1984*). Unless otherwise noted, ampicillin (Amp) (Sigma-Aldrich, MO, USA) was used at a concentration of 200 μg/mL, chloramphenicol (Cam) (Alfa Aesar, Heysham, UK) at 15 μg/mL, kanamycin (Kan) (Sigma-Aldrich) at 25 μg/mL, streptomycin (Sm) (Sigma-Aldrich) at 50 μg/mL and spectinomycin (Spec) (Sigma-Aldrich) at 20 μg/mL.

### Plasmid construction

All plasmids are listed in *Supplementary file 2*. To construct most plasmids, the desired gene or DNA fragment was amplified by PCR using appropriate DNA template and primers listed in *Supplementary file 3*. The amplified DNA fragment was digested with appropriate restriction enzymes (New England Biolabs, MA, USA) and ligated into the same restriction sites of the desired vector. NovaBlue cells were transformed with the ligation product and selected on LB plates containing appropriate antibiotics. MlaF and MlaB site substitution mutants were constructed by site-directed mutagenesis using the parent plasmids and primers listed in *Supplementary file 3*. The plasmid pET23/42(N) was used to construct pET23/42*His-mlaE* and pET23/42*His-mlaB.* To construct pET23/42(N), which contains a His-tag at the N-terminus, the first cloning site of pETDuet-1 (Novagen) was cloned and inserted into the pET23/42 plasmid (*Wu et al., 2006*). To construct pET22/42*mlaF(His-E)DCB, mlaFEDCB* was first amplified and inserted into NdeI/AvrII sites of the pET22/42 (*Chng et al., 2010b*). The coding sequence for the linker region-His$_6$ was subsequently inserted into pET22/42*mlaFEDCB* via two rounds of site-directed mutagenesis/insertion using 1-Linker-NHis-MlaE-N FWD/1-Linker-NHis-MlaE-C REV and 2-Linker-NHis-MlaE-N FWD/2-Linker-NHis-MlaE-C REV primer pairs. All constructs were verified by DNA sequencing (Axil Scientific, Singapore).

### Affinity purification experiments of His-tagged MlaF, MlaE, MlaD, and MlaB expressed at low levels

WT or mutant strains used in these experiments express the His-tagged Mla proteins from the pET23/42 plasmid under the control of the T7 promoter, which is transcribed inefficiently by endogenous non-T7 RNA polymerases. We believe that these proteins are produced at near physiological or low levels (as opposed to being over-expressed) because we have previously shown that leaky expression of MlaA from this plasmid is in fact lower than what is produced from the endogenous locus (*Chong et al., 2015*). For each strain, a 3-L culture (inoculated from an overnight culture at 1:100 dilution) was grown in LB broth at 37°C until OD$_{600}$ of ~0.5–0.7. Equal amounts of cells for each strain (normalized by OD$_{600}$) were harvested by centrifugation at 4,700 x *g* for 20 min. Cells were resuspended in 25 mL TBS containing 1 mM PMSF (Calbiochem, Darmstadt, Germany), 50 μg/mL DNase I (Sigma-Aldrich) and 100 μg/mL lysozyme (Calbiochem). Cells were passed once through a high pressure French Press (French Press G-M, Glen Mills, NJ, USA) homogenizer at 20,000 psi. Cell debris was removed by centrifugation at 4,700 x *g* for 10 min at 4°C. Subsequently, the supernatant was subjected to ultra-centrifugation (Model Optima L-100K, Beckman Coulter, CA, USA) at 145,000 x *g* for 1 hr at 4°C to separate membrane and soluble fractions. The membrane pellet fraction was extracted (5 mL of 50 mM Tris.HCl pH 8.0, 150 mM NaCl, 5 mM MgCl$_2$, 10% glycerol, 1%

*n*-dodecyl β-D-maltoside (DDM) (Merck Millipore, Italy) for 2 hr (with rocking at 4°C) and was subjected to second round ultra-centrifugation at 145,000 x *g* for 1 hr at 4°C. The supernatant was then incubated with 0.25 mL TALON cobalt resin (Takara Bio Inc, Japan), which had been equilibrated with 5 mL of wash buffer (50 mM Tris.HCl pH 8.0, 300 mM NaCl, 10% glycerol, 0.05% DDM, 10 mM imidazole), by rocking for 1 hr on ice. The mixture was later loaded onto a column and allowed to drain by gravity. The filtrate was passed through the resin again, drained and the column was washed with 5 x 2 mL of wash buffer and finally eluted with 1 mL of elution buffer (50 mM Tris.HCl pH 8.0, 150 mM NaCl, 10% glycerol, 0.05% DDM, 50 mM imidazole). The eluate was concentrated in an Amicon Ultra 10 kDa cut-off ultra-filtration device (Merck Millipore, Ireland) by centrifugation at 3,800 x *g* to ~100 µL. The concentrated sample was mixed with equal amounts of 2X Laemmli reducing buffer, either kept at room temperature (- heat) or boiled at 100°C for 10 min (+ heat), and subjected to SDS-PAGE and immunoblot analyses.

## Over-expression and purification of membrane complexes

MlaF(His-E)DB, MlaF(His-E), MlaF(His-E)D were over-expressed and purified from BL21(λDE3) cells harboring pET22/42*mlaF(His-E)CDB*, pET22/42*mlaF(His-E)* and pET22/42*mlaF(His-E)D*. In order to optimize amounts of MlaB during MlaF(His-E)DB purification, a second over-expression vector pCDF*mlaB* was introduced into BL21(λDE3) pET22/42*mlaF(His-E)CDB*. To over-express MlaF(His-E) B, pCDF*mlaB* together with pET22/42*mlaF(His-E)* were introduced into BL21(λDE3) cells. A 30-mL culture was grown from a single colony in LB broth supplemented with 200 µg/mL Amp and 50 µg/mL Sm (when necessary) at 37°C until $OD_{600}$ ~ 0.6. The cell culture was then used to inoculate a 3-L culture and grown at the same temperature until $OD_{600}$ ~ 0.6. For MlaF(His-E)DB and MlaF(His-E)B mutant complexes, 1 mM IPTG (Axil Scientific, Singapore) was added and the culture was grown at 37°C for another 3 hr. For other membrane sub-complexes, 0.1 mM IPTG was added and the culture was grown at 18°C for another 20 hr. Cells were pelleted by centrifugation at 4700 x *g* for 20 min and then resuspended in 25 mL TBS containing 1 mM PMSF (Calbiochem), 50 µg/mL DNase I (Sigma-Aldrich) and 100 µg/mL lysozyme (Calbiochem). Cells were passed once through a high pressure French Press (French Press G-M, Glen Mills) homogenizer at 20,000 psi. Cell debris was removed by centrifugation at 4700 x *g* for 10 min at 4°C. Subsequently, supernatant was subjected to ultra-centrifugation (Model Optima L-100K, Beckman Coulter) at 145,000 x *g* for 1 hr at 4°C to separate membrane and soluble fractions. The membrane pellet fraction was extracted (20 mL of 20 mM Tris.HCl pH 8.0, 300 mM NaCl, 5 mM $MgCl_2$, 10% glycerol, 1% *n*-dodecyl β-D-maltoside (DDM) (Merck Millipore) and subjected to second round ultra-centrifugation at 145,000 x *g* for 1 hr at 4°C. The supernatant was incubated with 1 mL TALON cobalt resin (Takara Bio Inc), which had been equilibrated with 20 mL of wash buffer (20 mM Tris.HCl pH 8.0, 300 mM NaCl, 5 mM $MgCl_2$, 10% glycerol, 0.05% DDM, 20 mM imidazole), by rocking for 1 hr on ice. The mixture was later loaded onto a column and allowed to drain by gravity. The filtrate was passed through the resin again, drained and the column was washed with 8 x 10 mL of wash buffer and eluted with 8 mL of elution buffer (20 mM Tris.HCl pH 8.0, 150 mM NaCl, 5 mM $MgCl_2$, 10% glycerol, 0.05% DDM, 50 mM imidazole for full complex and 200 mM imidazole for sub-complexes). The eluate was concentrated in an Amicon Ultra 10 kDa cut-off ultra-filtration device (Merck Millipore) by centrifugation at 3,800 x *g* to ~500 µL. Proteins were further purified by SEC system (AKTA, GE Healthcare, UK) at 4°C on a pre-packed Superdex 200 increase 10/300 GL column, using 20 mM Tris.HCl pH 8.0, 150 mM NaCl, 5 mM $MgCl_2$, 10% glycerol, 0.05% DDM as the eluent. For MlaF(His-E)B complexes, two columns were connected in series to allow better peak separation.

## Over-expression and purification of sMlaD-His and dLolB-His

sMlaD-His and dLolB-His were over-expressed and purified from BL21(λDE3) harboring pET22/42s*mlaD-His* and pET22b*lolB-His* (*Chng et al., 2010b*), respectively. A 5-mL culture was grown from a single colony in LB broth supplemented with 200 µg/mL Amp at 37°C until $OD_{600}$~ 0.6. The cell culture was then used to inoculate into 500 mL LB broth and grown at the same temperature until $OD_{600}$~ 0.6–0.7. At this time, 1 mM and 0.1 mM IPTG (Axil Scientific) were added to cultures and grown at 37°C for 3 hr and at 18°C for 20 hr to induce sMlaD-His and dLolB-His over-expression, respectively. Cells over-expressing sMlaD-His were pelleted by centrifugation at 4700 x *g* for 20 min and then resuspended in 20 mL TBS pH 6.25 (20 mM Tris.HCl pH 6.25, 300 mM NaCl) whereas cells

over-expressing dLolB-His were resuspended in 20 mL TBS pH 8.0 (20 mM Tris.HCl pH 8.0, 300 mM NaCl), both buffers containing 5 mM imidazole. We found that purification of sMlaD-His at ~ pH 6 gave more homogenous preparations. Buffers were supplemented with 1 mM PMSF (Calbiochem), 50 µg/mL DNase I (Sigma-Aldrich) and 100 µg/mL lysozyme (Calbiochem). Resuspended cells were passed twice through a high pressure French Press (French Press G-M, Glen Mills) homogenizer at 20,000 psi. Cell debris was removed by centrifugation at 4700 x $g$ for 10 min at 4°C. Subsequently, the supernatant was subjected to ultra-centrifugation (Model Optima L-100K, Beckman Coulter) at 145,000 x $g$ for 1 hr at 4°C to separate membrane and soluble fractions. The soluble fraction was incubated with 2 mL TALON cobalt resin (Takara Bio Inc) which had been equilibrated with 5 mL of TBS pH 6.25 (for sMlaD-His) and TBS pH 8.0 (for dLolB-His) both containing 20 mM imidazole, and incubated by rocking for 1 hr on ice. The resin mixtures were later loaded onto gravity column. The filtrates were collected and the columns were washed with 4 x 20 mL TBS pH 6.25 for sMlaD-His and TBS pH 8.0 for dLolB-His, both containing 20 mM imidazole. sMlaD-His and dLolB-His proteins were eluted from columns with 8 mL of 20 mM Tris.HCl pH 6.25, 150 mM NaCl, 200 mM imidazole and 20 mM Tris.HCl pH 8.0, 150 mM NaCl, 200 mM imidazole, respectively. The eluates were concentrated in an Amicon Ultra 10 kDa cut-off ultra-filtration device (Merck Millipore) by centrifugation at 3,800 x $g$ to ~500 µL. Proteins were further purified by SEC system (AKTA, GE Healthcare) at 4°C on a pre-packed Superdex 200 10/300 GL column, using 20 mM Tris.HCl pH 6.25, 150 mM NaCl (for sMlaD-His) and 20 mM Tris.HCl pH 8.0, 150 mM NaCl (for dLolB-His) as the eluents.

## SEC-MALS analysis to determine absolute molar masses of sMlaD-His and MlaF(His-E)DB

Prior to each SEC-MALS analysis, a preparative SEC was performed for bovine serum albumin (BSA) (Sigma-Aldrich) to separate monodisperse monomeric peak and to use as a quality control for the MALS detectors. In each experiment, monomeric BSA was injected before the protein of interest and the settings (calibration constant for TREOS detector, Wyatt Technology) that gave the well-characterized molar mass of BSA (66.4 kDa) were used for the molar mass calculation of the protein of interest.

For sMlaD-His, Superdex 200 Increase 10/300 GL column was equilibrated with 20 mM Tris.HCl pH 6.25, 150 mM NaCl. The injected protein concentration was kept at 5 mg/mL. Light scattering and refractive index data were collected online using miniDAWN TREOS (Wyatt Technology, CA, USA) and Optilab T-rEX (Wyatt Technology, CA, USA), respectively, and analyzed by ASTRA 6.1.5.22 software (Wyatt Technology). The dn/dc value was set to 0.185 mL/g for the soluble protein (*Slotboom et al., 2008*).

For MlaF(His-E)DB, SEC column equilibration and BSA calibration were performed in 20 mM Tris. HCl pH 8.0, 150 mM NaCl, 5 mM MgCl₂, 10% glycerol and 0.05% DDM. 1 mg/mL complex was used and data collected as above. To calculate non-proteinaceous part of the complex, we used the protein-conjugate analysis in ASTRA software. In this analysis, we used dn/dc value of 0.143 mL/g and 0.185 mL/g for DDM and protein complex, respectively (*Slotboom et al., 2008*). For BSA, UV extinction coefficient of 0.66 mL/(mg.cm) was used. For the MlaF(His-E)DB complex, that was calculated to be 0.84 mL/(mg.cm), based on its predicted stoichiometric ratio $MlaF_2(His_6-E)_2D_6B_2$.

## Lipid extraction and thin layer chromatography

To analyze endogenously bound PLs, lipids from purified sMlaD-His, dLolB-His (*Chng et al., 2010b*), BL21(λDE3) and EH150 cell pellet were extracted according to Bligh-Dyer method (*Bligh and Dyer, 1959*). For BL21(λDE3) lipid extraction, an overnight BL21(λDE3) culture was grown in 5 mL LB broth at 37°C. 1.5 mL of the culture was pelleted down and resuspended in 100 µL deionized water. For the lipid extraction from temperature-sensitive *E. coli* EH150 strain (*Hawrot and Kennedy, 1975*), a 3-mL culture (inoculated from an overnight culture at 1:100 dilution) was grown first at 30°C until OD₆₀₀ of ~0.5–0.7 and subsequently at 42°C for another 4 hr. 1.5 mL culture was taken, pelleted down and resuspended in 100 µL deionized water. 2.5 mg of purified proteins (sMlaD-His and dLolB-His) were used for PL extraction.

Purified protein solutions, BL21(λDE3) and EH150 pellet resuspensions were mixed with 3.75 volumes of chloroform:methanol (1:2 vol/vol). The mixtures were vortexed and sonicated sequentially for 30 s for three times. The mixtures were centrifuged at 21,000 x $g$ for 5 min and the supernatants

were recovered. Appropriate volumes of chloroform and TBS (for protein samples) or deionized water (for cell pellet) were added to the supernatants to form a two phase mixture chloroform:methanol:water (2:2:1.8). The mixtures were then centrifuged at 4000 x $g$ for 5 min to separate organic and aqueous phases. Organic phase was gently removed to another vial and the organic solvent was evaporated under $N_2$ gas. Remaining dried lipids were dissolved in 50 µL chloroform:methanol (4:1 vol/vol) and 10 µL spotted onto a TLC Silica gel 60 $F_{254}$ plate (Merck). The plate was developed in chloroform:methanol:water (65:25:4) solvent system, left to dry at room temperature and then stained with iodine vapor for 5 min. Visualization was done with G:Box Chemi-XX 6 (Genesys version 1.4.3.0, Syngene).

## $^{31}$P Nuclear magnetic resonance (NMR) analysis for extracted phospholipid species

To find out the selectivity of sMlaD-His for different PL head groups, we performed $^{31}$P NMR analysis with Bruker AV500 (500 MHz) instrument (MA, USA). Commercial *E. coli* phosphatidylethanolamine (PE), phosphatidylglycerol (PG) and cardiolipin (CL) (Avanti Polar Lipids, AL, USA) were used as standards. Synthetic 1,2-dimyristoyl-sn-glycero-3-phosphate (sodium salt) (14:0) (Avanti Polar Lipids) was used as phosphatidic acid (PA) standard. Phosphatidylserine (PS) was extracted from *E. coli* EH150 strain by accumulating PS at 42°C (*Hawrot and Kennedy, 1975*). To achieve resolved $^{31}$P signals, phospholipid-Triton X-100 micelles were prepared with some modifications (*London and Feigenson, 1979*; *Sotirhos et al., 1986*). PE, PG, CL and PA were first solubilized in chloroform: methanol (4:1) at 3 mM, 2 mM, 4 mM and 1 mM final concentrations, respectively. Lipids from sMlaD-His (from 10 mg of purified protein), BL21(λDE3) and EH150 strains were extracted as described above (*Bligh and Dyer, 1959*). Organic solvents from samples were evaporated under $N_2$ gas and remaining dried lipids were resuspended in 1 mL water containing 5% Triton X-100 (w/v) and 10% $D_2O$ (v/v), and subsequently sonicated for 1 hr at room temperature. NMR analysis was performed at room temperature and acquisition times for all samples were 10 hr ~ 40,000 number of scans (NS), except PA sample for 3 hr. Fourier transformation of FID files and integration of phosphate peak areas were done in MestReNova 9.0 software.

## Native mass spectrometry analysis

Purified protein samples were desalted with Amicon Ultra-0.5 mL 10 kDa cut-off centrifugal filters (Merck Millipore) to exchange the initial buffer with 20 mM ammonium acetate pH 6.8 and to eliminate eventually co-purified and loosely bound contaminating small molecules. Non-denaturing analyses were performed in positive ion mode. The sample (6 µL, 20 µM) was directly injected at a flow of 2 µL/min into a quadrupole time-of-flight (QTOF) 6550 Agilent mass spectrometer equipped with an Agilent 1200 HPLC system (CA, USA). The nanoelectrospray voltage was set to 1500 V (Vcap), temperature 280°C, drying gas flow 11 L/min and fragmentor 175 V. The samples were injected in 50 mM ammonium acetate pH 6.8 and spectra acquired at a scan rate of 1 spectra/sec. Data were collected over a mass range of 1000–8000 m/z. Denaturing mass spectrometry, to confirm the theoretical mass of the monomeric purified protein, was performed injecting the samples in positive ion mode using conditions that could break the protein-ligand complex (protein diluted 1:10 in 0.2% formic acid in 50% acetonitrile). Data were acquired between 100 and 8000 m/z (positive mode) to monitor the protein conformational states. The sample was analyzed with the same mass spectrometer connected with the Agilent 1200 series ChipLC system and using a C8 chip (Zorbax 300SB, Agilent). The following solvents were used for the protein analysis on the C8 reversed phase HPLC: 0.2% formic acid in water (solvent A), 0.2% formic acid in acetonitrile (solvent B). The 6550 QTOF electrospray voltage was set to 1600 V (Vcap), temperature 200°C, drying gas 14 L/min, fragmentor voltage 175 V. The denatured protein analysis data were collected in MS only mode and deconvoluted using Bioconfirm software (Agilent) and the Maximum Entropy deconvolution algorithm.

## Enzyme-coupled ATPase assay

ATP hydrolytic activity was determined using an NADH enzyme-linked assay (*Nørby, 1988*) adapted for a microplate reader (*Kiianitsa et al., 2003*). 50-µL reactions contained assay buffer (20 mM Tris. HCl pH 8.0, 150 mM NaCl, 5 mM $MgCl_2$ 10% glycerol, 0.05% DDM) with 200 µM NADH (Sigma-Aldrich), 20 U/mL lactic dehydrogenase (Sigma-Aldrich), 100 U/mL pyruvate kinase (Sigma-Aldrich),

0.5 mM phosphoenolpyruvate (Alfa Aesar) and different ATP (Sigma-Aldrich) concentrations. The assays were performed at room temperature and fluorescence emission at 340 nm was measured using a SPECTRAmax 250 microplate spectrophotometer equipped with SOFTmax PRO software (Molecular Devices, CA, USA). Readings were taken in ~9 s intervals. The rate of decrease of NADH fluorescence (due to oxidation) was calculated from a linear fit to each 10 min time course and converted to ATP hydrolysis rates with a standard curve obtained using known ADP concentrations (*Figure 5—source data 1*). Samples were performed in technical triplicates and data were fit to the built-in Hill equation in Origin 9.

## OM permeability studies

OM sensitivity against SDS/EDTA was judged by colony-forming-unit (CFU) analyses on LB agar plates containing indicated concentrations of SDS/EDTA. Briefly, 5-mL cultures were grown (inoculated with overnight cultures at 1:100 dilution) in LB broth at 37°C until $OD_{600}$ reached ~0.5–0.7. Cells were normalized according to $OD_{600}$, first diluted to $OD_{600}$ = 0.1 (~$10^8$ cells), and then serially diluted (ten-fold) in LB using 96-well microtiter plates. 2 µL of the diluted cultures were manually spotted onto the plates, dried, and incubated overnight at 37°C. Plate images were visualized by G: Box Chemi-XT4 (Genesys version 1.4.3.0, Syngene).

## SDS-PAGE, immunoblotting and staining

All samples subjected to SDS-PAGE were mixed with equal amounts of 2X Laemmli reducing buffer. The samples were subsequently either kept at room temperature (- heat) or subjected to boiling at 100°C for 10 min (+ heat). Equal volumes of the samples were loaded onto the gels. Unless otherwise stated, SDS-PAGE was performed according to Laemmli using the 4–20% Tris.HCl gradient gels (*Laemmli, 1970*). After SDS-PAGE, gels were visualized by either Coomassie Blue staining (Sigma-Aldrich), silver staining (Silver Quest, Invitrogen) or subjected to immunoblot analysis.

Immunoblot analysis was performed by transferring protein bands from the gels onto polyvinylidene fluoride (PVDF) membranes (Immun-Blot 0.2 µm, Bio-Rad, CA, USA) using semi-dry electroblotting system (Trans-Blot Turbo Transfer System, Bio-Rad). Membranes were blocked by 1X casein blocking buffer (Sigma-Aldrich). α-His antibody (pentahistidine) conjugated to the horseradish peroxidase (HRP) (Qiagen, Hilden, Germany) was used at a dilution of 1:5,000. Luminata Forte Western HRP Substrate (Merck Millipore) was used to develop the membranes and chemiluminescence signals were visualized by G:Box Chemi-XT4 (Genesys version 1.4.3.0, Syngene).

# Acknowledgements

We thank Zhi-Soon Chong (NUS) for providing knock-out strains and initial data on complementation studies. We also thank Rahul Shrivastava (NUS) and Jie Ren Tan (NUS High School) for providing the pET22/*42smlaD-His* and preliminary purification data. Finally, we acknowledge Ross Tomaino (Taplin MS Facility, Harvard Medical School) for help with tandem MS sequencing. BE was supported by the SINGA scholarship. This work was supported by the National University of Singapore Start-up funding, the Singapore Ministry of Education Academic Research Fund Tier 1 and Tier 2 (MOE2013-T2-1-148) grants (to S-SC), and by grants from the National University of Singapore Life Sciences Institute (LSI), the Singapore National Research Foundation (NRFI2015-05) and a BMRC-SERC joint grant (BMRC-SERC 112 148 0006) from the Agency for Science, Technology and Research (A*STAR) Singapore (to MRW).

# Additional information

## Funding

| Funder | Grant reference number | Author |
| --- | --- | --- |
| Singapore International Graduate Award | Scholarship | Bilge Ercan |
| National University of Singapore | Start-up funding | Shu-Sin Chng |

| | | |
|---|---|---|
| Ministry of Education - Singapore | Academic Research Fund Tier 1 | Shu-Sin Chng |
| Ministry of Education - Singapore | Academic Research fund Tier 2 (MOE2013-T2-1-148) | Shu-Sin Chng |
| National University of Singapore | Life Science Institute | Markus R Wenk |
| National Research Foundation Singapore | NRFI2015-05 | Markus R Wenk |
| Agency for Science, Technology and Research | BMRC-SERC 112 148 0006 | Markus R Wenk |

The funders had no role in study design, data collection and interpretation, or the decision to submit the work for publication.

### Author contributions
ST, BE, Conception and design, Acquisition of data, Analysis and interpretation of data, Drafting or revising the article; FT, Contributed native protein MS analysis data, Conception and design, Acquisition of data, Analysis and interpretation of data, Drafting or revising the article; ZYF, HYAW, Acquisition of data, Analysis and interpretation of data; MRW, S-SC, Conception and design, Analysis and interpretation of data, Drafting or revising the article

### Author ORCIDs
Shu-Sin Chng, http://orcid.org/0000-0001-5466-7183

## Additional files

### Supplementary files
• Supplementary file 1. Bacteria strains used in this study.

• Supplementary file 2. Plasmids used in this study.

• Supplementary file 3. Primers used in this study.

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
