## [Decision Letter]

Thank you for submitting your article "Defining key roles for auxiliary proteins in an ABC transporter that maintains bacterial outer membrane lipid asymmetry" for consideration by *eLife*. Your article has been favorably evaluated by Richard Losick as the Senior editor and four reviewers, one of whom is a member of our Board of Reviewing Editors and another is Jue Chen (Reviewer #3).

The reviewers have discussed the reviews with one another and the Reviewing Editor has drafted this decision to help you prepare a revised submission.

Summary:

The Mla system maintains the lipid asymmetry in the outer membrane (OM) of *Escherichia coli*. It has been proposed that Mla transports excess phospholipids from the OM to the inner membrane (IM) to keep the outer leaflet of the OM free of phospholipids. Here, Thong et al. present the first biochemical characterization of the Mla IM complex. They demonstrate that MlaBDEF form a stable ABC transporter complex and they study the function of accessory proteins MlaBD. They show that the periplasmic domain of MlaD forms a hexamer that binds phospholipids, the proposed substrates of the Mla system. They demonstrate that MlaD decreases the ATPase activity of the MlaBEF complex, possibly to prevent futile ATP hydrolysis in the absence of substrate. They also show that MlaB is necessary for both the association of the ATPase MlaF with its transmembrane partner MlaF and the ATPase activity of the ABC transporter.

Three of the reviewers think the detailed characterization of Mla system components described in this manuscript has been missing from the field and is therefore a much-needed contribution. Nevertheless, major questions about the system remained unanswered, despite the authors having the tools to answer them, and one reviewer thought the contribution was of limited scope with a knowledge gap between the data and the proposed model. The authors openly discuss the larger questions such as are PLs inserted into the inner or outer leaflet of the IM? Are PLs only handed unidirectionally from MlaC to MlaD? Is ATPase activity critical for this directionality? The Mla system is interesting because of its novel PL transport activity – this paper addresses essential basic auxiliary questions of the system (do all of the IM proteins interact? Does it hydrolyze ATP?), but does not answer any of the major transport questions sampled above. Given that transport of phospholipids between the IM and OM is the least understood essential process in Gram-negative envelope biogenesis, the Reviewing Editor agrees that overall the study is "highly influential". However, the inclusion of additional experiments that address the larger questions raised above, though not essential, would strengthen the paper.

Essential revisions:

1) The authors report that a *mlaB* null has no Mla function based on sensitivity to SDS/EDTA. Supporting this, their biochemical data show that MlaB is needed for both the association of MlaF with MlaE and the ATPase activity of the complex. However, Malinverni and Silhavy reported that the *mlaB* mutant has significantly milder defects than the other *mla* mutants. The authors should explain this discrepancy.

2) The authors report that the presence of MlaD affects cooperativity of the binding for ATP. Where is the data showing this?

3) The authors assume that in the detergent micelles, the phospholipids bound to MlaD come off (Discussion, second paragraph). From this assumption, they propose that MlaD lowers the ATPase activity of the Mla complex when not bound to substrate and that this activity increases upon substrate binding. Can they show that MlaD was not bound to phospholipids? If they cannot, they should then tone down the statement.

4) The pull-down experiments are properly controlled, however there is a surprisingly high level of non-specific binding. At times it is difficult to see the bands that are being indicated by the authors. The authors should comment on this.

5) It is not clear if the assays used to detect PG and PE bound to MlaD are qualitative or quantitative. A specific, quantitative PL competition assay to see if MlaD preferentially binds PG over PE, and to what degree seems more appropriate to draw a conclusion about PL preference (as opposed to relying on relative PL concentrations in the cell). Entire papers focus on such experiments, and the conclusions drawn here are not backed by thorough quantitative evidence. Therefore, the text must be altered here as the authors have not provided convincing evidence for PL preference. The likelihood that an inner membrane protein like MlaD naturally associates with PLs despite it's like role in PL transport should also be addressed in the text.

---

## [Author Response]

*Three of the reviewers think the detailed characterization of Mla system components described in this manuscript has been missing from the field and is therefore a much-needed contribution. Nevertheless, major questions about the system remained unanswered, despite the authors having the tools to answer them, and one reviewer thought the contribution was of limited scope with a knowledge gap between the data and the proposed model. The authors openly discuss the larger questions such as are PLs inserted into the inner or outer leaflet of the IM? Are PLs only handed unidirectionally from MlaC to MlaD? Is ATPase activity critical for this directionality? The Mla system is interesting because of its novel PL transport activity – this paper addresses essential basic auxiliary questions of the system (do all of the IM proteins interact? Does it hydrolyze ATP?), but does not answer any of the major transport questions sampled above. Given that transport of phospholipids between the IM and OM is the least understood essential process in Gram-negative envelope biogenesis, the Reviewing Editor agrees that overall the study is "highly influential". However, the inclusion of additional experiments that address the larger questions raised above, though not essential, would strengthen the paper.*

We agree with the reviewers that additional experiments that address the interesting aspects of how the OmpC-Mla system transports PLs would strengthen the paper. In this manuscript, we have provided the first detailed characterization of the MlaFEDB complex, and hence, established a platform to begin answering questions on PL transport. While studies to gain mechanistic insights into the process of PL transport mediated by the Mla proteins are extremely valuable, and will be our next focus, however, we think that these studies are beyond the scope of this manuscript.

*Essential revisions:*

*1) The authors report that a mlaB null has no Mla function based on sensitivity to SDS/EDTA. Supporting this, their biochemical data show that MlaB is needed for both the association of MlaF with MlaE and the ATPase activity of the complex. However, Malinverni and Silhavy reported that the mlaB mutant has significantly milder defects than the other mla mutants. The authors should explain this discrepancy.*

We have repeatedly assayed the SDS/EDTA sensitivity of the ∆*mlaB* strain and found no significant differences compared to other *mla* mutants, and we agree with the reviewer that this result is internally consistent with our biochemical data. We do not have a good explanation for why the ∆*mlaB* mutant has milder defects than other *mla* mutants in the Silhavy paper. However, we note that we have constructed our ∆*mlaB* strain in almost the same manner as that paper. The only difference is that we have not removed the Cam^R^ cassette in our strain, which might have given rise to the subtle differences observed. We have not modified the text so as not to distract the readers.

*2) The authors report that the presence of MlaD affects cooperativity of the binding for ATP. Where is the data showing this?*

We have based our interpretations of cooperativity in ATP binding on the Hill coefficient values derived upon fitting our data. We have shown that ATP binding for the MlaFEB complex exhibits cooperativity (Hill coefficient = 1.5 ± 0.5) while that for the MlaFEDB complex does not (Hill coefficient = 1.0 ± 0.6). We have given this information in the Figure 5 legend but failed to include it (for MlaFEDB) in the text. We have now added this in the appropriate location.

*3) The authors assume that in the detergent micelles, the phospholipids bound to MlaD come off (Discussion, second paragraph). From this assumption, they propose that MlaD lowers the ATPase activity of the Mla complex when not bound to substrate and that this activity increases upon substrate binding. Can they show that MlaD was not bound to phospholipids? If they cannot, they should then tone down the statement.*

We do not think that it will be straightforward to distinguish between specific and non-specific PL binding in the MlaFEDB complex since membrane proteins are likely to be associated at least with annular PLs. We are therefore not able to show that MlaD within the complex was not bound to PLs. We have now altered the text to tone down our initial statement.

*4) The pull-down experiments are properly controlled, however there is a surprisingly high level of non-specific binding. At times it is difficult to see the bands that are being indicated by the authors. The authors should comment on this.*

We believe that several factors contribute to the high level of non-specific binding in our pull-down experiments.

1) We have expressed the His-tagged Mla proteins at low levels in order to preserve native interactions and reduce false positives. These proteins are expressed from the pET23/42 plasmid under the control of the T7 promoter, which is transcribed inefficiently by endogenous non-T7 RNA polymerases. We have previously shown that expression of MlaA from this plasmid is in fact lower than what is produced from the endogenous locus (Chong et al., 2015). Having lower levels of His-tagged proteins means that fewer sites on the affinity column would be occupied, thus leading to higher non-specific binding. Because of the low amounts of His-tagged proteins, we also have to use much higher volumes of bacterial cultures for our experiments, which also lead to higher non-specific binding.

2) Our pull-down experiments involve membrane proteins, which are inherently hydrophobic and hence more prone to non-specific binding to the affinity column, even in the presence of detergents.

3) We have used the ability of the His-tag epitope to bind to a cobalt affinity resin as the means for affinity purification. This makes the experiment more amenable to larger scale purification. In contrast to immunoprecipitation, however, the His-tag-cobalt interaction is weaker, thereby giving rise to more non-specific binding.

We have not modified the text to reflect these points so as not to distract the readers.

*5) It is not clear if the assays used to detect PG and PE bound to MlaD are qualitative or quantitative. A specific, quantitative PL competition assay to see if MlaD preferentially binds PG over PE, and to what degree seems more appropriate to draw a conclusion about PL preference (as opposed to relying on relative PL concentrations in the cell). Entire papers focus on such experiments, and the conclusions drawn here are not backed by thorough quantitative evidence. Therefore, the text must be altered here as the authors have not provided convincing evidence for PL preference. The likelihood that an inner membrane protein like MlaD naturally associates with PLs despite it's like role in PL transport should also be addressed in the text.*

The techniques we have used are either qualitative (TLC) or semi-quantitative (^31^P NMR). The latter technique has been used by others to determine the relative compositions of different PL species in lipid mixtures (Sotirhos et al., 1986). However, we agree with the reviewers that we have not provided convincing evidence for PL preference of MlaD for transport. We have now modified the text to reflect this.

We agree that an inner membrane protein may naturally associate with PLs. However, in our experiments to look at PL binding to MlaD, we have used only the soluble domain of the protein. While this domain is tethered to the membrane via the transmembrane helix in the native form of the protein, it is entirely soluble and does not associate with membranes when expressed and purified on its own. We therefore believe that bound PLs in soluble MlaD directly reflect its role in PL transport. We have modified the text to clarify this point.